# Growth Hormone Secretagogue Receptor (GHSR) Is Elevated in Myocardial Tissues of DMD *mdx:utrn*^−/−^ Mice, and Correlates Strongly with Inflammatory Markers, and Negatively with Cardiac Function

**DOI:** 10.3390/cells14131002

**Published:** 2025-07-01

**Authors:** Maedeh Naghibosadat, Andrew McClennan, Margarita Egiian, Reema Flynn-Rizk, Tyler Lalonde, Carlie Charron, Anish Chhabra, Leonard G. Luyt, Savita Dhanvantari, Lisa M. Hoffman

**Affiliations:** 1Department of Pathology and Laboratory Medicine, University of Western Ontario, London, ON N6A 3K7, Canada; maya.naghibosadat@mail.utoronto.ca (M.N.); sdhanvan@lawsonimaging.ca (S.D.); 2Imaging Program, Lawson Research Institute/SJHC, London, ON N6C 2R5, Canada; andrew.mcclennan@cuanschutz.edu (A.M.); megiian@uwo.ca (M.E.); rflynnri@uwo.ca (R.F.-R.); achhabr6@uwo.ca (A.C.); 3Department of Medical Biophysics, University of Western Ontario, London, ON N6A 3K7, Canada; 4Department of Chemistry, University of Western Ontario, London, ON N6A 3K7, Canada; tyler.lalonde@muhc.mcgill.ca (T.L.); carlie.charron@dal.ca (C.C.); lluyt@uwo.ca (L.G.L.); 5Department of Anatomy & Cell Biology, University of Western Ontario, London, ON N6A 3K7, Canada

**Keywords:** dilated cardiomyopathy, Duchenne muscular dystrophy, inflammation, GHSR

## Abstract

Dilated cardiomyopathy affects greater than 1 in 2500 patients worldwide, including those with the neuromuscular disorder Duchenne muscular dystrophy (DMD). While inflammation within skeletal muscle is strongly associated with DMD pathology, the key biomarkers for inflammation and possible targets for therapy within cardiac tissue in DMD-associated dilated cardiomyopathy remain to be identified. One such potential target is the myocardial ghrelin-growth hormone secretagogue receptor (GHSR) system, which is associated with cardiomyocyte survival and inhibition of inflammation. We sought to determine alterations in myocardial GHSR together with markers of cardiac inflammation using *mdx:utrn*^−/−^ mice as a model for DMD-associated dilated cardiomyopathy. With traditional histopathology, we determined that the pathology of DMD in *mdx:utrn*^−/−^ mice was characterized by disruption of myofiber organization, lymphocytic infiltration, and extensive cardiomyocyte vacuolization and necrosis surrounding areas of fibrosis in the left ventricular wall and apex. Using a fluorescent ghrelin analog, Cy5-ghrelin (1–19), to visualize GHSR with fluorescence confocal microscopy, we demonstrate that GHSR is elevated in *mdx/utrn*^−/−^ myocardial tissues and correlates strongly with both F4-80 (activated macrophages) and IL-6 (pro-inflammatory cytokine), and negatively with cardiac function. We also show that GHSR can be visualized in pro-inflammatory macrophages, suggesting a direct role for GHSR in the inflammatory progression of DMD.

## 1. Introduction

Dilated cardiomyopathy (DCM) represents the third leading cause of heart failure with a worldwide prevalence estimated to be greater than 1 in 2500–3000 [1,2,3]. DCM etiology is extremely heterogeneous, with genetic inheritance arising in ~50% of patients [1,2,3], including those as young as 8–10 years of age afflicted with the neuromuscular disorder Duchenne muscular dystrophy (DMD) [4]. Thus, there is a critical need to more clearly understand the mechanisms of pathogenesis in order to identify therapeutic targets for intervention.

Inflammation is a key component of DMD skeletal muscle pathology [5,6] and has largely been demonstrated in the *mdx* mouse model of DMD [7,8,9]. There is also evidence of progressive inflammation in cardiac tissue; at 6 months of age, there is elevated F4/80 antibody immunofluorescence, a marker of macrophage infiltration [10], and at 9 months of age, *mdx* mouse hearts are fibrotic, hypertrophied, and exhibit both diminished contractility [11] and tachycardia due to increased sympathetic and decreased parasympathetic activity [12]. However, while *mdx* mice are widely used as a model of DMD, the pathological features in these mice are relatively mild compared with human DMD, particularly regarding DCM [13,14,15]. In contrast to the *mdx* mouse, the *mdx:utrn*^−/−^ mouse lacks both dystrophin and utrophin, and the absence of both genes accelerates the development of DMD. At an early age (3–4 weeks), the *mdx:utrn*^−/−^ mouse develops severe and progressive skeletal muscle pathology, spinal curvature, and limited mobility [16,17,18], which more closely mirrors the human condition in which these symptoms are typically evident by approximately 8–10 years of age [19,20]. Importantly, at ~8–10 weeks of age, *mdx:utrn*^−/−^ mice also develop a dilative cardiac pathology akin to DMD and other DCM patients, including cardiomyocyte membrane damage that leads to ischemia, inflammation, and fibrosis [16,17,18]. Our group has used 3D echocardiography to demonstrate age-related decreases in left ventricular ejection fraction (LVEF) and cardiomyocyte hypertrophy in *mdx:utrn*^−/−^ mice [21], thus supporting the use of these mice at a later stage of life as a clinically relevant model of DMD-associated cardiomyopathy.

In addition to the immune phenotyping of DCM in DMD, investigations into other possible therapeutic targets have provided additional insights into pathophysiology. One such potential target is the growth hormone secretagogue receptor (GHSR), a 7-transmembrane, G protein-coupled receptor, and its natural ligand, ghrelin, a 28-amino-acid peptide. Ghrelin is synthesized in cardiomyocytes and acts through myocardial GHSR to promote cardiomyocyte survival and contractility, and to inhibit apoptosis, fibrosis, and inflammation [22]. These properties of ghrelin have led to studies investigating its use as a therapy for several heart conditions, and one study showed that ghrelin administration significantly improved heart function and decreased LV remodeling and fibrosis in a mouse model of dilated cardiomyopathy in which cardiomyopathy occurs through a knock-in of the deletion mutation deltaK210 in cardiac troponin [23]. Interestingly, administration of ghrelin in *mdx* mice improves muscle function by inhibiting inflammasome activation [24], thus indicating a role for ghrelin-GHSR signaling in the inhibition of inflammation.

The use of ghrelin or other GHSR agonists as potential therapeutics for cardiomyopathies highlights the need to determine the dynamics of the receptor to predict outcomes of such therapies. Our group has developed novel fluorescent analogs of GHSR ligands to reveal changes in the dynamics of myocardial GHSR during cardiomyocyte differentiation [25], in a mouse model of diabetic cardiomyopathy [26], and in human myocardial tissue before and after cardiac transplantation [27] and heart disease [28]. Therefore, changes in the distribution of myocardial GHSR appear to be associated with cardiac pathology in both mouse models and in human heart disease.

In the present study, we sought to determine alterations in myocardial GHSR together with markers of cardiac inflammation in DMD DCM using *mdx:utrn*^−/−^ mice as a model. We used a combination of traditional histopathology and confocal imaging using a novel fluorescent ghrelin analog, Cy5-ghrelin (1–19), to examine both pathological changes in cardiac tissue architecture and GHSR and inflammatory markers in the progression of DCM.

## 2. Materials and Methods

### 2.1. Mouse Housing and Breeding

All mice were housed at the Lawson Health Research Institute Animal Care Facility in controlled conditions (19–23 °C, 12 h light/dark cycles). Food and water were provided ad libitum. The animal protocol (#2008-067) was approved by the Western University Institutional Animal Use Subcommittee and conducted according to guidelines set by the Canadian Council on Animal Care (CCAC).

As a negative control, wild-type C57BL/6 mice were purchased from Charles River Laboratories (Wilmington, MA, USA). Heterozygous *mdx* mice (*mdx:utrn*^+/−^, a generous gift from Dr. Robert Grange, Virginia Polytechnic Institute and State University, Blacksburg, VA, USA) were crossed to generate *mdx:utrn*^−/−^ mice, which lack both utrophin and dystrophin. Only males of wild-type and *mdx:utrn*^−/−^ mice at the age of 15–17 weeks were used in this study. At this age range, *mdx:utrn*^−/−^ mice demonstrate the severe phenotype of DMD disease 16–18 and do not tend to live past 20 weeks of age. In this study, the DMD mouse refers exclusively to the *mdx:utrn*^−/−^ mouse.

### 2.2. Animal Genotype Validation

For genotyping, tail snips or ear-notches were collected, and DNA was extracted according to the standard PCR genotyping protocol. Briefly, tail snips or ear-notches were lysed using Proteinase K at 55 °C overnight, followed by PCR amplification of the utrophin gene [29]. The sets of primers that were used to confirm the presence of utrophin were 5′- TGCCAAGTTCTAATTCCATCAGAAGCTG -3′ (forward primer) and 5′- CTGAGTCAAACAGCTTGGAAGCCTCC-3′ (reverse primer). Gene deletion was confirmed by the visualization of a band at 310 kb on a 1% agarose gel containing ethidium bromide.

### 2.3. Cardiac Tissue Preparation

All hearts were obtained from mice from a previous study in which cardiomyopathy had been measured by determining left ventricular ejection fraction through micro-CT and 3D echocardiography [21]. In that study, the hearts from each of three 15–17-week-old WT and three *mdx:utrn*^−/−^ mice were removed immediately after CO_2_ euthanasia and fixed in 10% buffered formalin for 48 h [21]. Hearts were then cut axially into two equal sections and embedded in paraffin. Tissue processing and sectioning were conducted by the Pathology Core Facility at Robarts Research Institute (Western University, London, ON, Canada). Tissue blocks were sliced into 5 μm serial sections, with a total of 40 sections for each block of cardiac tissue. To ensure that sections were representative of the whole cardiac tissue, serial sections were taken every 10 slices for histological analysis and fluorescent staining. The selected tissue sections (5 per heart) were deparaffinized and dehydrated with a series of xylene, 100%, 90% and 70% ethanol prior to staining, as we have conducted previously [30].

### 2.4. Histological Imaging

Hematoxylin and Eosin (H&E) and Masson’s Trichrome staining were performed by the Pathology Core Facility at Robarts to qualitatively assess tissue architecture and the level of fibrosis, respectively. Images were acquired on a Zeiss Axioscope microscope with Northern Eclipse software v7.0 (Empix Imaging Inc., Mississauga, ON, Canada) under 10×, 20×, and 40× objectives. Non-overlapping fields of view were taken to image the entire tissue for each section with the 10× objective. As well, the entire tissue section was raster scanned at 10× to provide an unbiased view of the staining. To have more insight into each section, 6 fields of view were taken with the 20× and 40× objectives.

### 2.5. Fluorescence Immunostaining

Heat-mediated antigen retrieval was performed with 10 mM sodium citrate buffer in Phosphate-Buffered Saline (PBS), pH = 7.4, for 20 min prior to staining. Tissue sections were incubated with blocking buffer (10% Donkey serum Cat no. D9663, MilliporeSigma Canada Ltd., Oakville, ON, Canada) + 1% bovine serum albumin (BSA) in PBS) for 1 h at room temperature. Slides were then incubated with primary antibodies (Table 1) overnight at 4 °C, followed by incubation with appropriate secondary antibodies (Table 1) for 2 h at room temperature. Finally, Cy5-ghrelin (1–19) (1:100) was added to slides and incubated for 30 min, as described previously [25,26,27,28].

### 2.6. Confocal Microscopy

#### 2.6.1. Fluorescence Image Acquisition

Sections were imaged on a Nikon A1 R confocal microscope equipped with NIS Elements software (version 6.10.01) using a 60× oil-immersion objective. The microscope was equipped with four lasers: DAPI (405 nm), FITC (490–519 nm), TRITC (561–594 nm), and Cy5 (633–647 nm), and they were switched on separately to reduce the crosstalk between the fluorochromes. Five sections were imaged per cardiac tissue, and for each section, five regions of interest (ROIs) were captured. For high-resolution imaging of the microvasculature, the Nyquist option was used to capture the field of interest, and then image acquisition was conducted with a higher pixel resolution (1024 × 1024), as we have conducted previously [31,32]. To display the maximum contrast and brightness, images were deconvoluted using the 2D deconvolution algorithm on NIS Elements [31,32]. This method reduces chromatic aberration, enhances the true signal, and reduces the background.

#### 2.6.2. Fluorescence Image Analysis

The analysis was conducted using the ROI statistic option of NIS Elements software. Fluorescence intensity analysis was conducted on the 5 ROIs from each section, and for immunofluorescence images, the intensity was normalized to the negative control (secondary antibody alone), as described above. The fluorescence intensity analysis was conducted for Cy5-ghrelin (1–19), F4-80, and IL-6. The extent of colocalization was determined by Pearson correlation coefficient values extracted using NIS Elements software, as we have conducted previously [28,31,32].

#### 2.6.3. H&E and Masson’s Trichrome Image Acquisition

Tissue sections were imaged on a Nikon inverted spinning disk confocal equipped with NIS Elements software using a 20× objective lens. Each whole tissue section was scanned at high resolution to create a single, large 6 mm × 6 mm image of the cardiac tissue. This was achieved by capturing overlapping images, or tiles, which were then combined using the Nikon Dimensions (ND) Large-Image-Stitching algorithm. A 15% overlap was applied through the binding system to ensure seamless automatic stitching of the tiles.

#### 2.6.4. H&E and Masson’s Trichrome Image Analysis

Image analysis was conducted using Media Cybernetics’ Image Pro AI software (v. 11.1). Two custom protocols were created based on pretrained models as follows: For Masson’s Trichrome analysis, six images of entire heart sections were analyzed per sample. The percentage of collagen was calculated as the ratio of the blue-stained area to the total area of the heart section. For H&E analysis, five non-overlapping ROIs were selected to represent the cell nuclei distribution accurately. Three sections were analyzed per sample, and values were averaged to obtain a single measurement.

### 2.7. Statistical Analyses

For GHSR analysis, statistical analyses were conducted, and graphs were plotted using GraphPad Prism v.7.03 (San Diego, CA, USA). A T-test was used to compare the levels of GHSR between the DMD and age-matched WT using SPSS software v.24.0. A *p*-value of <0.05 was considered significant. To correlate the level of inflammation with GHSR, linear regression analysis was run on the F4-80 vs. GHSR data and also on IL-6 vs. GHSR data. To determine the association between LVEF and GHSR, linear regression analysis was used on previously published LVEF values on these mice [21] and Cy5-ghrelin fluorescence intensities.

For H&E and Masson’s Trichrome, statistical analysis was conducted in R (version 4.3.2). The Mann–Whitney U-test was used to compare the groups. A *p*-value < 0.05 was considered significant. Cliff’s Delta was calculated to quantify the effect size.

## 3. Results

### 3.1. Characterization of Cardiac Tissue Pathology in the DMD Heart

The extent of tissue pathology and fibrosis in wild-type and DMD (*mdx:utrn*^−/−^) mice was investigated using hematoxylin and eosin (H&E) and Masson’s trichrome. Our analysis focused on the left ventricular myocardium, which is the primary region affected by dilated cardiomyopathy. We first compared the cellular organization between WT and DMD mice at 15–17 weeks of age (Figure 1). In WT mice, cardiac muscle fibers were organized, and cardiomyocytes were either mononucleated or binucleated with a central nucleus in the extensive pinkish-red cytoplasm (Figure 1C). In contrast, cardiac myofibers in DMD mice were shrunken, had lost their nuclei, and eosin staining was pale (Figure 1F). Vacuoles were also found within the cardiomyocyte sarcoplasm and could be a manifestation of myofiber necrosis and cardiomyopathy (Figure 1E). Pyknotic nuclei and aggregation of lymphocytes indicate necrosis and chronic inflammation within the myocardium of the DMD mice (Figure 1E). Our findings illustrate that the pathology of dilated cardiomyopathy in DMD mice includes pyknosis (black arrow), lymphocytic infiltration (green arrow), and vacuolization (red arrow). A larger number of cell nuclei was present in the hearts of DMD mice compared to the wild type, suggesting a leukocyte infiltration.

To determine the extent of both cardiac tissue degeneration and fibrosis simultaneously, we aligned the H&E and Masson’s trichrome staining in the same area of the cardiac tissue. In WT mice, cardiac myofibers in the lateral wall of the LV were uniformly organized and had centrally located nuclei (Figure 2C) and were negative for collagen staining (Figure 2A). In contrast, in tissues from DMD mice, there was a significant amount of fibrosis in the lateral wall of the left ventricle in DMD mice (Figure 2B and Figure 3D), with more peripherally located nuclei in the cardiomyocytes (Figure 2D). There were pyknotic nuclei and lymphocyte infiltration above the collagen fibers (Figure 3C,F).

The extent of fibrosis in the LV apex of DMD mice was also determined (Figure 4). At 10X magnification (Figure 4A,D), the highly disorganized architecture of the myofibers was visualized, and there was a large extent of collagen and fat deposition in this area (Figure 4A,D). At a higher magnification, pyknotic nuclei and aggregation of lymphocytes were seen (Figure 4C,F), indicating cardiomyocyte necrosis and chronic inflammation, respectively. There were also large areas of adipose tissue and collagen deposition that surrounded the disorganized myofibers (Figure 4F).

While there appeared to be increased leukocyte infiltration and collagen content in cardiac tissue from DMD (*mdx:utrn*^−/−^) mice, quantification of percent collagen content and number of nuclei showed no statistically significant differences (Figure 5). However, Cliff’s delta of 0.56 (number of nuclei) and 0.78 (percent collagen content) suggests a large effect size.

### 3.2. Localization of GHSR in Cardiac Tissue

Our lab has previously shown that the dynamics of myocardial GHSR are altered in end-stage heart failure and heart disease in humans [27,28] and in a mouse model of diabetic cardiomyopathy [26]. Therefore, we reasoned that the expression and distribution of GHSR would also be altered during the progression of dilated cardiomyopathy in DMD mice. Specifically, we investigated the association of GHSR with the myocardium and cardiac microvasculature, heart function, and the inflammatory processes that characterize DCM in DMD.

Cardiac tissues from WT and DMD mice were stained with Cy5-ghrelin (1–19) along with AlexaFluor488-isolectin to evaluate the presence of GHSR in the microvasculature within the myocardium. GHSR was barely detectable in WT myocardium and was significantly elevated (*p* < 0.0001) in DMD myocardium (Figure 6A, middle column). Microvessels in the myocardium of WT mice demonstrated an organized and cylindrical shape; in contrast, in DMD mice, the cylindrical architecture of the microvessels in the left ventricle appeared to be disrupted, and in some cases, contained structures strongly positive for isolectin (Figure 6A, left column). The overall fluorescence intensity of Cy5-ghrelin (1–19) significantly increased in the myocardial tissue of DMD mice, as visualized in Figure 6A, middle column, and as quantified in Figure 6B.

Closer examination of the merged fluorescence image in DMD tissue did not indicate any potential overlap between Cy5-ghrelin (1–19) and Alexa488-isolectin within the microvessels (Figure 6A, white arrows). However, there appeared to be co-localization in the structures within the microvessels (Figure 6A, yellow arrows), suggesting that GHSR may be localized in these structures in addition to cardiomyocytes.

### 3.3. Correlation Between LVEF and GHSR

Our previous results have shown a negative correlation between GHSR and left ventricular ejection fraction (LVEF) in patients who underwent cardiac transplantation [27]. We therefore wished to determine the relationship between GHSR and LVEF in *mdx:utrn*^−/−^ DMD mice. In a previous study, we showed that LVEF significantly decreased in DMD mice by 15 weeks of age [21]. We performed linear regression analysis on these previously reported LVEF values and Cy5-ghrelin (1–19) fluorescence intensities in tissues from the same WT and DMD mice. There was a strong and significant inverse correlation between LVEF and myocardial GHSR expression (* *p* < 0.05, r = −0.863). (Figure 7).

### 3.4. The Relationship Between GHSR and Cardiac Inflammation in DMD

Due to the highly inflammatory environment of DCM in DMD mice, as we showed in Figure 1, Figure 2, Figure 3 and Figure 4, we investigated the relationship between GHSR and inflammation using two markers: F4-80, which binds to activated macrophages, and interleukin-6 (IL-6), a general indicator of inflammation. In WT myocardium, there was very low Cy5-ghrelin (1–19) binding, no detectable expression of F4-80, and regularly shaped microvessels, as indicated by AlexaFluor488-isolectin fluorescence (Figure 8A, top row). In contrast, in the myocardium of DMD mice, there were markedly higher levels of F4-80 and Cy5-ghrelin (1–19) binding and disruption of the microvessel architecture (Figure 8A, bottom row). When total fluorescence intensities were quantified, there were significant elevations of both Cy5-ghrelin (1–19) and F4-80 in DMD mice (Figure 8B). Levels of F4-80 and Cy5-ghrelin (1–19) showed a significant positive correlation (R^2^ = 0.97, *p* < 0.004, Figure 8B).

As in Figure 6A, there appeared to be discrete structures within microvessels that showed co-localization of Cy5-ghrelin (1–19) and AlexaFluor488-isolectin, and also with F4-80 (yellow arrows). We employed high-resolution imaging to more closely investigate this co-localization (Figure 9). The fluorescent signals from AlexaFluor488-isolectin and F4-80 overlapped within discrete areas; therefore, AlexaFluor488-isolectin also labeled macrophages in the DMD myocardium. As well, there was overlap with Cy5-ghrelin (1–19), suggesting that GHSR may be expressed in the cardiac macrophages in DMD myocardium (Figure 9). We calculated a Pearson’s correlation coefficient of 0.926 between GHSR and F4-80, thus suggesting the presence of GHSR on activated macrophages.

As with F4-80, there was a significant increase in IL-6 in DMD myocardium that correlated with the increase in GHSR (Figure 10). Interestingly, IL-6 fluorescence was also observed within the lumen of the microvessels in the DMD mice (Figure 10A) indicated by the white arrow) and did not appear to co-localize with Cy5-ghrelin (1–19). Quantification of fluorescence intensities showed that IL-6 expression levels were significantly greater in DMD myocardium and showed a significant correlation (*** *p* < 0.002, R^2^ =0.9286) with GHSR (Figure 10B).

## 4. Discussion

In this study, we provide a detailed examination of DMD DCM cardiac pathology, with a focus on inflammation. Specifically, H&E staining detailed the extent of chronic inflammation in cardiac tissues in *mdx:utrn*^−/−^ mice at 15–17 weeks of age. It showed the existence of lymphocyte aggregates in the lateral wall and apex of the left ventricle, which is a hallmark of chronic inflammation. We further demonstrated that expression of macrophage marker F4-80 and the pro-inflammatory cytokine IL-6 are elevated in DMD cardiac tissue relative to healthy wild-type cardiac tissue. We demonstrate that GHSR is also elevated in *mdx/utrn*^−/−^ myocardial tissues and correlates strongly with both F4-80 and IL-6 expression, and negatively with cardiac function as measured by left ventricular ejection fraction. One potentially novel finding is the presence of GHSR on activated macrophages but not monocytes, thus strengthening the association of GHSR with inflammation in DCM.

The findings of this study importantly support earlier studies demonstrating that membrane damage in cardiomyocytes, cellular necrosis, and fibrosis are present in *mdx/utrn*^−/−^ cardiac muscle [17,18]. Additionally, the pathology of DCM is exacerbated by chronic inflammation, an area that has been considerably less studied. Recent studies have characterized resident immune cells in healthy mouse skeletal muscle and then quantified differences in skeletal muscle inflammation between two dystrophic mouse models: *mdx* and *mdx/utrn*^+/−^ [7]. Other studies have demonstrated that a chronic inflammatory response dominates the skeletal muscle molecular signal in mdx mice [8]. The few studies that have assessed inflammation in DMD DCM have utilized the weakly affected *mdx* mouse [10].

Fluorescence immunostaining using F4-80 as a macrophage marker was used to detect the existence of macrophages in the DMD myocardium. In chronic inflammation, both M1 (pro-inflammatory) and M2 (regeneration) macrophages are present [9]. However, F4-80 cannot differentiate between M1 and M2 macrophages. To evaluate the macrophage phenotypic distribution, future studies should assess CD11c for M1 and CD206 as a marker of M2 [33]. Macrophages implement a range of important immunoregulatory and inflammatory functions by secreting chemokines and cytokines and also by degrading the extracellular matrix that surrounds cardiomyocytes. Damaged cardiomyocytes activate pro-inflammatory signaling pathways through NF-kB, which upregulates the production of pro-inflammatory cytokines such as TNF-α, IL-1, and IL-6. The present study demonstrates an increase in the expression of the cytokine IL-6, a marker of the monocyte-macrophage transition [34] in the cardiac tissue, and also within the microvasculature of DMD mice. These results are in agreement with the literature, which shows high levels of IL-6 in the serum and inflammatory infiltrates of DMD patients. There is evidence that IL-6 is also elevated in the diaphragm muscle of *mdx* mice, which is the most affected muscle in this strain of DMD mouse [35,36,37,38]. In the heart, IL-6 has dual roles in both cardioprotection and cardiac failure. In acute inflammation, IL-6 participates in neurogenesis and wound healing and maintains cardiac integrity. In contrast, in chronic inflammation, IL-6 activates constant signaling through the NF-kB pathway, which leads to fibrosis [38]. Our results suggest that the elevated levels of IL-6 are due to chronic inflammation that will eventually lead to heart failure.

Having characterized inflammatory changes in dilated cardiomyopathy in the *mdx:utrn*^−/−^ mouse model, we next investigated whether myocardial GHSR levels are altered in *mdx:utrn*^−/−^ mice, as assessed by the binding of the imaging probe Cy5-ghrelin (1–19). Using this fluorescent peptide analog of ghrelin, we have previously shown that GHSR levels are altered before and after cardiac transplantation in humans [27] and also in diabetic cardiomyopathy in mice [26]. Our results demonstrate an elevation in GHSR in the myocardium of the *mdx:utrn*^−/−^ mice at end-stage cardiomyopathy, which is in agreement with the elevated levels of GHSR reported in end-stage human heart failure [27]. However, these findings are in contrast with mouse diabetic cardiomyopathy studies, where the level of GHSR is down-regulated using both fluorescence microscopy with Cy5-ghrelin (1–19) and Western blot analysis [26]. The reason for this discrepancy between the mice with diabetic cardiomyopathy and the *mdx:utrn*^−/−^ mice could possibly be explained by the fact that they were in different stages of cardiac disease. The mouse model of diabetic cardiomyopathy showed very mild changes in heart function and no signs of inflammation or fibrosis (26). In contrast, the *mdx:utrn*^−/−^ mice had late-stage cardiomyopathy, with a highly fibrotic and inflammatory pathology. Therefore, we suggest that GHSR may be elevated in late-stage cardiomyopathy and down-regulated in early-stage cardiomyopathy in mice. A limitation of this study is the lack of data at early stages of DMD in the *mdx:utrn*^−/−^ mice, which would have provided a more precise readout of the temporal changes in cardiac GHSR during the progression of DMD.

This study has shown that GHSR is expressed in DMD murine macrophages but not monocytes in the myocardium, suggesting a possible mechanism for its association with the inflammatory phenotype of DCM in DMD. There is evidence that GHSR is expressed on immune cells, and in macrophages in particular, indicating that ghrelin can have direct effects on the inflammatory processes associated with metabolism and aging through GHSR signaling [39,40]. In aged mice, the ablation of GHSR reduced the macrophage infiltrates and also promoted the macrophage phenotypical shift towards M2 (anti-inflammatory phase) [41]. As well, GHSR is present and elevated in alveolar macrophages during sepsis-induced acute respiratory distress syndrome in rats, and ghrelin administration [42] inhibited signaling through the Wnt/β-catenin pathway, reducing apoptosis and inflammation. These studies support our findings and provide evidence that GHSR signaling can ameliorate inflammation associated with a variety of chronic diseases.

Some limitations of our study are that a GHSR agonist was not administered, and that we provide correlational data. This was a retrospective study on mice from a previous study, and the goal was to characterize changes in GHSR in inflammatory cardiac pathology [30]. As well, we used a small number of mice in this study. However, we have shown similar patterns of correlation between GHSR and heart function in two studies utilizing human tissue [27,28] and in a recent study in a canine model of heart failure [43] (Sullivan et al., 2024). Future studies will examine the therapeutic use of GHSR agonists in the treatment of DCM in DMD.

Despite the advancements in imaging and biomarkers research for cardiomyopathy, distinguishing early pathogenesis and understanding disease mechanisms remain unsolved. Innovations in early recognition of cardiomyopathy can improve the survival and quality of life for patients and reduce the disease progression and hospital admission rate. These advancements can also open new doors for targeted therapies. Imaging GHSR in *mdx:utrn*^−/−^ mice represents a step towards the detection of early biomarkers for inflammation and dilated cardiomyopathy.

## Figures and Tables

**Figure 1 cells-14-01002-f001:**
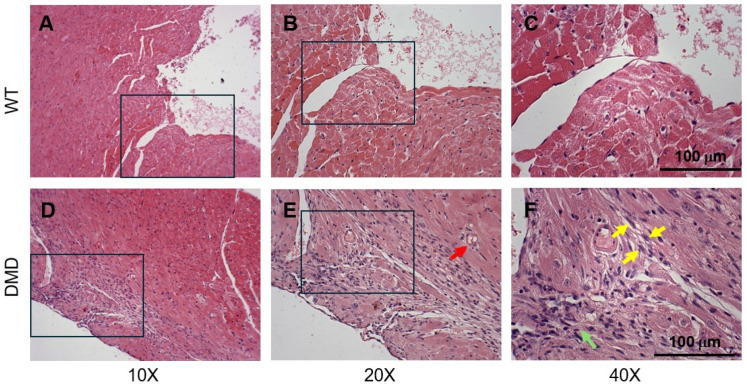
Representative images of cardiac tissue architecture of the left ventricle in wild-type and DMD (*mdx:utrn*^−/−^) mice at 15–17 weeks of age (*n* = 3). Cardiac tissue sections were stained with Hematoxylin and Eosin (H&E), and images were acquired at 10×, 20×, and 40× magnification in wildtype (WT, top row) and DMD (bottom row) mice. (**A**,**B**), Cardiac muscle fibers form an interconnecting three-dimensional network. (**C**), Cardiomyocytes contain one or two nuclei, which are centrally located with extensive eosinophilic cytoplasm. (**D**,**E**), Cardiac myofibers in DMD mice are loose. The red arrow indicates cardiomyocyte vacuolation, which is a marker of cardiac tissue degeneration. (**F**), The sizes of nuclei in the cardiomyocytes are variable. Yellow and green arrows indicate fibroblast and lymphocyte aggregates, respectively. Boxes represent magnified areas.

**Figure 2 cells-14-01002-f002:**
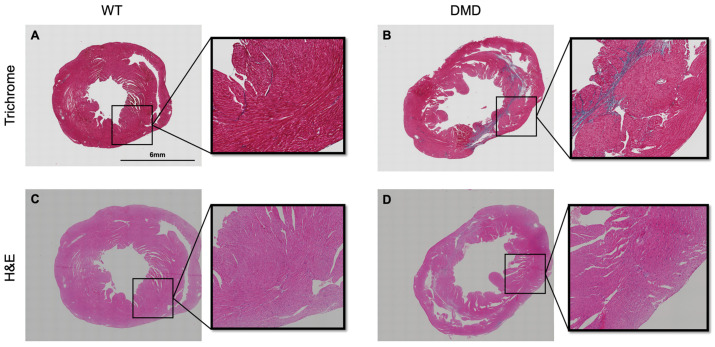
Representative histology images of whole cardiac tissue in both wild-type (*n* = 3) and DMD (*mdx:utrn*^−/−^) mice (*n* = 3). (**A**,**B**) Masson’s Trichrome staining in WT and DMD mice. The cardiac muscle is bright red, and there is no collagen content (blue staining) in the wild-type myocardium. Cardiac tissue in DMD mice exhibits increased collagen deposition (fibrosis), mainly concentrated around the left ventricle. (**C**,**D**) H&E staining in WT and DMD mice. In wild-type mice, myocardial fibers are organized. The cardiomyocytes are either mononucleated or binucleated and centrally located, whereas in DMD mice, they are more peripherally located. Boxes indicate the magnified areas.

**Figure 3 cells-14-01002-f003:**
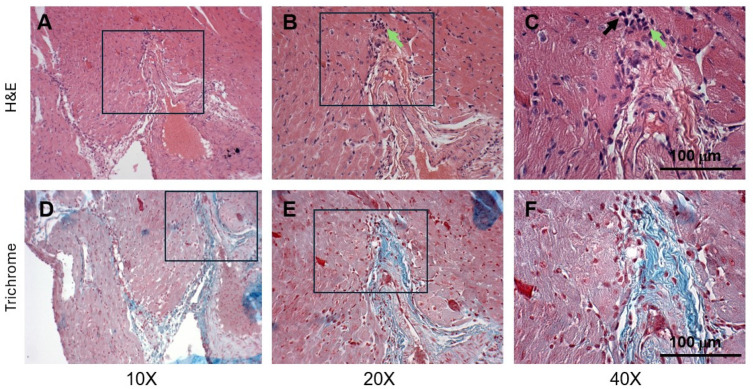
Representative histology images of the left ventricle in DMD (*mdx:utrn*^−/−^) mice (*n* = 3). (**A**–**C**) H&E staining at 10×, 20×, and 40× magnification, respectively. Myocardial fibers are disorganized. The cardiomyocytes’ nuclei are acentric and variable in size. The black arrow indicates an area of pyknotic nuclei, and the green arrow points to lymphocytes. (**D**–**F**) Masson’s trichrome staining; the panels align with the H&E staining above. (**D**) The diffuse pattern of collagen deposition in the DMD heart. (**F**) The size of nuclei in the cardiomyocytes is variable. Boxes represent the magnified areas. Panels A and D are not completely aligned due to the blue artifacts in Masson’s Trichrome staining.

**Figure 4 cells-14-01002-f004:**
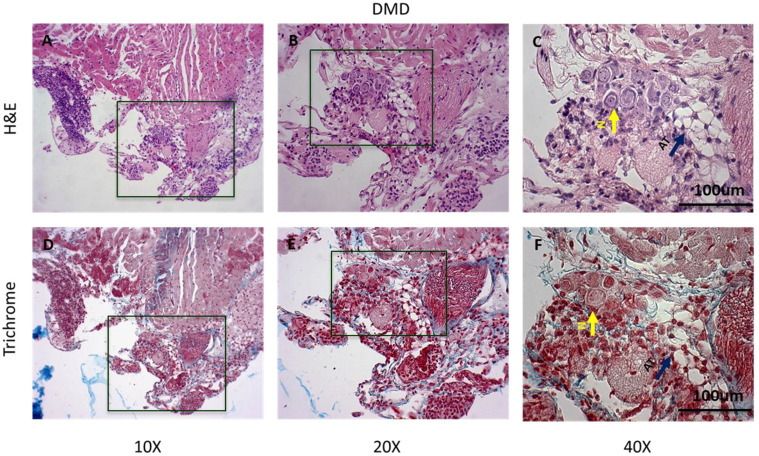
Representative histology images of the left ventricular apex in DMD (*mdx:utrn*^−/−^) mice (*n* = 3). (**A**–**C**) H&E staining at 10×, 20×, and 40× magnification, respectively. High magnification in panel C reveals the disruption in myocardial tissue organization, leukocyte infiltration, adipose tissue deposition (blue arrow), and neurons (yellow arrow). (**D**–**F**) Masson’s trichrome staining. These panels align with the H&E staining above. (**E**) There is diffuse fibrosis (blue), and cardiac tissue stains pale red. (**F**) The blue arrow indicates the adipose tissue in the myocardium, and the adjacent myofibers are degenerated. (**F**) Yellow and blue arrows indicate neurons and adipose tissue, respectively. Boxes represent the magnified areas.

**Figure 5 cells-14-01002-f005:**
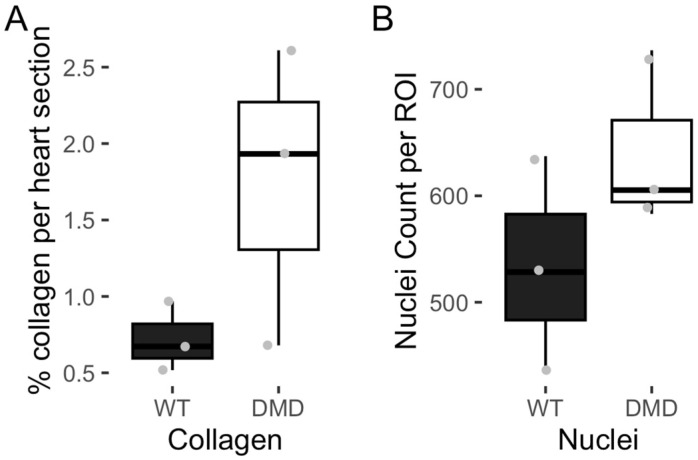
(**A**) Collagen content in cardiac tissue of DMD and WT mice (*n* = 3, *p* = 0.2, Cliff’s delta = 0.78 (95% CI: [−0.18, 0.98])). (**B**) Nuclei Count in cardiac tissue of DMD and WT mice (*n* = 3, *p* = 0.4, Cliff’s delta = 0.56 (95% CI: [−0.61, 0.96])). Data are represented as mean ± SEM. Each point represents an average per biological replicate.

**Figure 6 cells-14-01002-f006:**
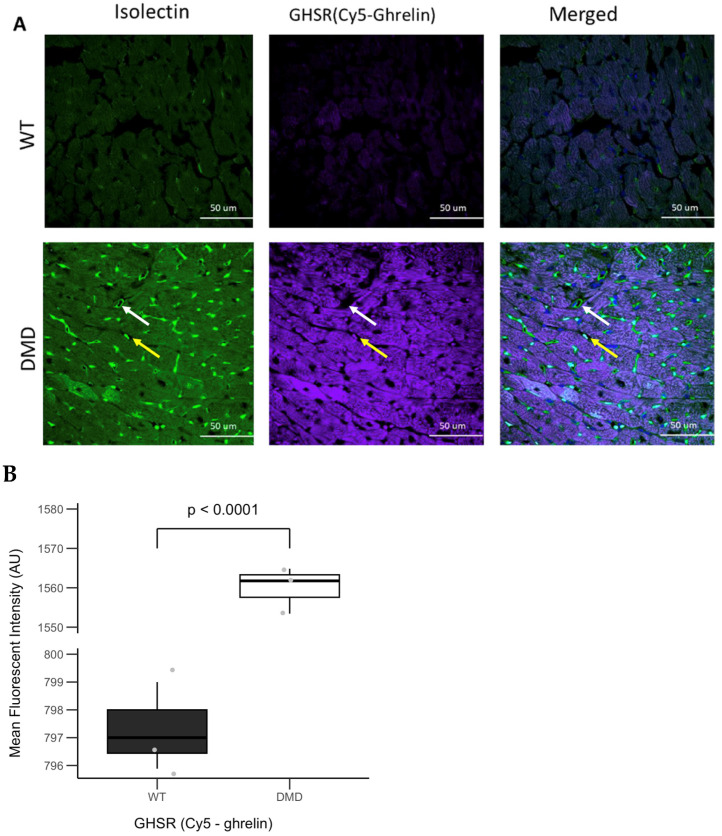
GHSR expression is elevated in DMD cardiac tissue. (**A**) Representative fluorescent images of the left ventricle in wild-type and DMD mice (*mdx:utrn*^−/−^). The cardiac tissues were stained with Cy5-ghrelin (1–19) and isolectin to show GHSR (purple) and microvessels (green), respectively. Nuclei were visualized with DAPI (blue). Cardiac microvasculature in the DMD heart looks aberrant and has an irregular shape, whereas in wild type, microvasculature has the normal circular shape. (**B**) GHSR fluorescence intensity is elevated in DMD mice (*n* = 3, *p* < 0.0001). Data are represented as mean fluorescence intensity ± SEM. Each point represents the average of fluorescence intensities per mouse.

**Figure 7 cells-14-01002-f007:**
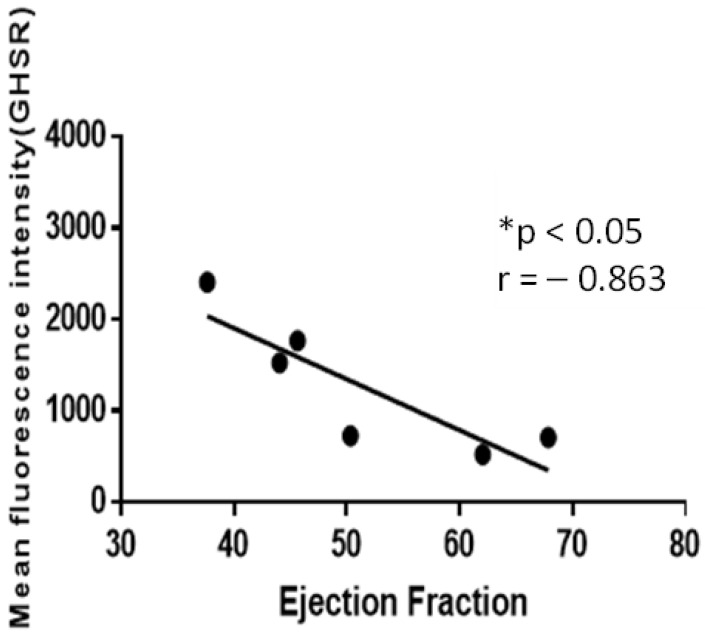
Correlation between ejection fraction (%) and GHSR in cardiac tissues in WT and DMD mice. There was a significant inverse correlation between the ejection fraction and Cy5-ghrelin (1–19) fluorescence intensity in the DMD cardiac tissue. Each data point represents values from one mouse.

**Figure 8 cells-14-01002-f008:**
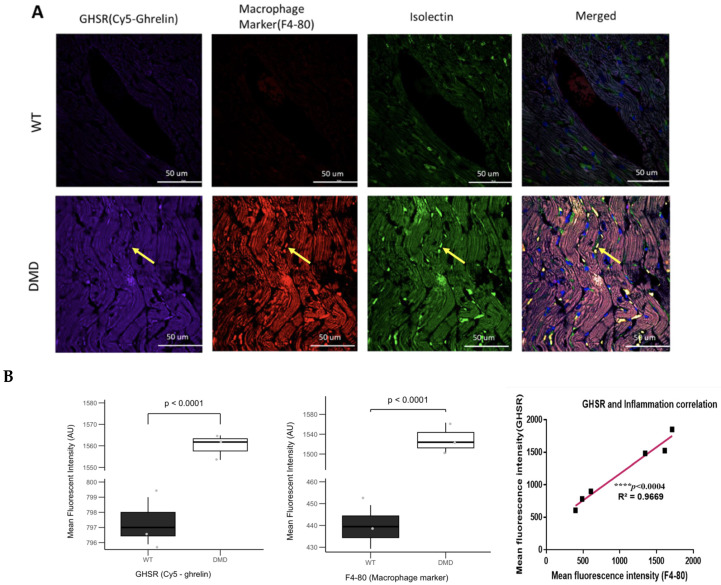
GHSR colocalizes with the macrophage marker F4-80. (**A**) Representative fluorescent images of the left ventricle in wild-type and DMD (*mdx:utrn*^−/−^) mice show Cy5-ghrelin (1–19), F4-80, and isolectin staining to visualize GHSR (purple), macrophages (red), and the vasculature (green), respectively. Nuclei are visualized with DAPI. Yellow arrows indicate structures within microvessels that are positive for GHSR, F4-80 and isolectin. (**B**) Quantification of fluorescence intensities for GHSR (*n* = 3, *p* < 0.0001) and F4-80 (*n* = 3, *p* < 0.0001). Data are represented as mean fluorescent intensity ± SEM. Each point represents the average of fluorescence intensities per mouse. Linear regression analysis indicates a strong correlation between GHSR and F4-80 (**** *p* < 0.0004, R^2^ = 0.9669).

**Figure 9 cells-14-01002-f009:**
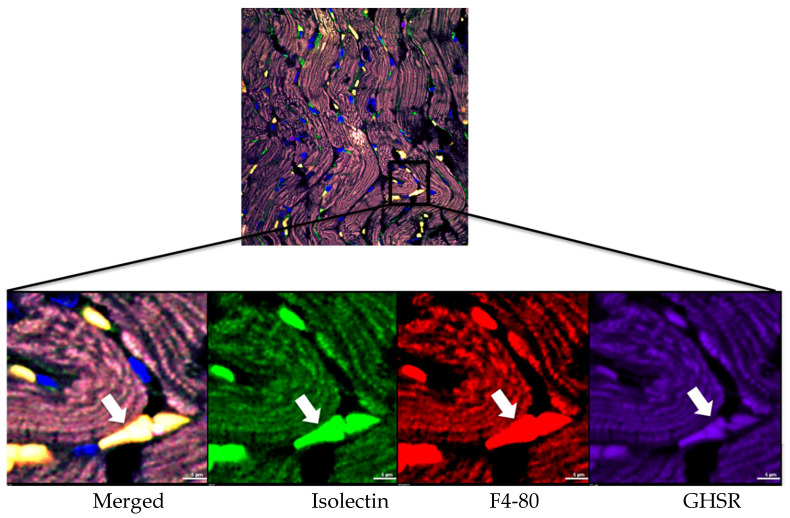
GHSR is expressed in cardiac macrophages. The ROI indicated in the top panel was subject to Nyquist analysis to increase the resolution. Cardiac tissue from DMD mice was stained with Cy5-ghrelin (1–19), F4-80, and isolectin to visualize the GHSR (purple), macrophages (red), and the vasculature (green), respectively. Nuclei were visualized with DAPI. White color indicates the colocalization of the three signals (red, green, and purple), which indicates the presence of GHSR in cardiac macrophages (arrows).

**Figure 10 cells-14-01002-f010:**
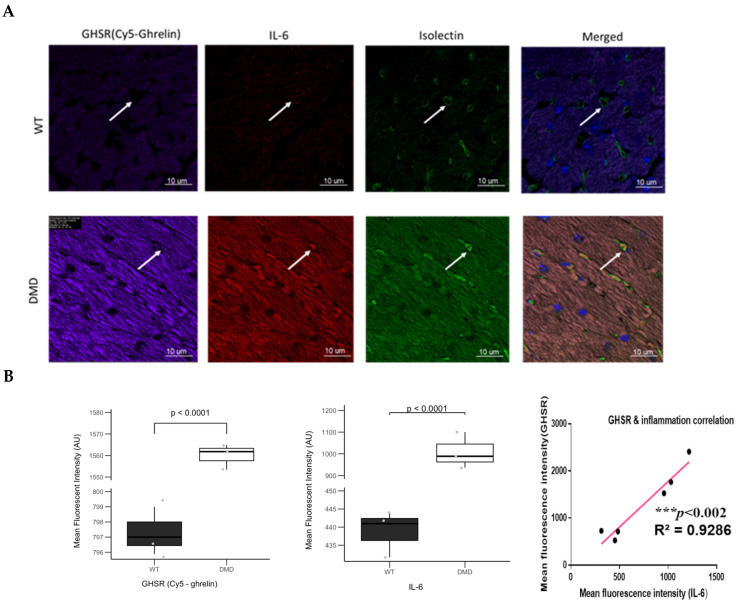
GHSR and the pro-inflammatory cytokine IL-6 are increased in cardiac tissue from DMD mice. (**A**) Representative confocal fluorescence images of the left ventricular myocardium in WT and DMD (*mdx:utrn*^−/−^) mice. Cardiac tissues were stained with Cy5-ghrelin (1–19), IL-6, and AlexaFluor488-isolectin to visualize GHSR, IL-6 cytokine, and the microvessels, respectively. Nuclei were visualized with DAPI. The arrows indicate the cardiac microvessels of WT mice (top row) and IL-6 staining within the microvessels in DMD myocardium (bottom row). (**B**) Quantification of fluorescence intensities for GHSR (*n* = 3, *p* < 0.0001) and IL-6 (*n* = 3, *p* < 0.0001). Data are represented as mean fluorescent intensity ± SEM; each point represents the average of fluorescence intensities per mouse. Linear regression analysis indicates a strong positive correlation for the GHSR vs. IL-6 in the cardiac tissues (*** *p* < 0.002, R^2^ = 0.9286).

**Table 1 cells-14-01002-t001:** Antibodies used for immunofluorescence microscopy.

Antibody	Source	Dilution	Target	Catalog	Research Resource Identification Number
Rat F4-80	Abcam	1:200	F4-80 glycoprotein on murine macrophages	ab16911	AB-443548
Mouse IL-6	Abcam	1:100	IL-6 Cytokine	ab9324	AB-307175
Rabbit GHSR	SantaCruz Biotechnology	1:500	GHSR	sc-374515	AB-10987651
Isolectin GS-IB4 Alexa Fluor 488 Conjugate	Life Technologies	1:100	Endothelial cells and macrophages	I21411	AB-2314662
Alexa Fluor-594 conjugated donkey anti- Rat IgG	Life Technologies	1:500	Rat IgG	A21209	AB-2535795
Alexa Fluor-594 conjugated donkey anti-mouse IgG	Life Technologies	1:500	Mouse IgG	A21203	AB-141633

## Data Availability

The original contributions presented in this study are included in the article. Further inquiries can be directed to the corresponding author.

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
