# Peer review of "Growth Hormone Secretagogue Receptor (GHSR) Is Elevated in Myocardial Tissues of DMD mdx:utrn−/− Mice, and Correlates Strongly with Inflammatory Markers, and Negatively with Cardiac Function"

_cells, 2025, doi:10.3390/cells14131002_

Round 1
Reviewer 1 Report (Previous Reviewer 1)
Comments and Suggestions for Authors
The authors have significantly improved the presentation of the work and responded adequately to comments.
Author Response
We thank the reviewer for their helpful comments in improving the manuscript.
Reviewer 2 Report (Previous Reviewer 2)
Comments and Suggestions for Authors
I agree with the authors that western blotting is not a foolproof technique for quantization of protein amounts. Like every technique western blot has drawbacks and advantages and so does quantification using microscopy. I understand proper controls for background subtraction have been used but that does not address issues with comparison of two images with different background. Moreover, relying solely on a single technique, such as microscopy, without corroborating evidence from complementary methods, introduces uncertainties regarding the robustness of the findings.
Author Response
please see the attachment

Reviewer 3 Report (New Reviewer)
Comments and Suggestions for Authors
In the paper “Growth hormone secretagogue receptor (GHSR) is elevated in myocardial tissues of DMD mdx:utrn-/- mice, and correlates strongly with inflammatory markers, and negatively with cardiac function” Naghibosadat et al, try to find a correlation between GHSR expression and the progression of dilative cardiomyophathy in a dystrophic mouse model. They use several techniques to characterize inflammation and GHSR expression both in WT mice and in dystrophic ones. One of the biggest limitation of the study is represented by sample size as extensively stated by the author. The study is very interesting and could enhance the diagnostic and early detection of cardiac myopathy. In light of these considerations, there are some issues that needs to be clarified:
Major comments:
- H&E is for sure a powerful and predictive staining to carefully analyse muscle morphology however, at the magnification showed in the paper it is very difficult to assess clearly what the author indicates in the results, such as loss of striation or distinction with fibroblast. If the authors have higher and clearer magnification of those images, it would be easier.
- Masson Trichrome on the WT sections does not seem right as not even nuclear staining is clear; moreover, how the author distinguish with this staining the fat deposition? Usually Sirius red staining is used to assess it. In addition, it seems quite strange the total absence of any type of matrix/fat/fibrotic tissue in the WT counterpart.
- Since mice were collected at 15-17 weeks of age, could it be that the pathology was already too much extensive? The authors state that the mice die around 20 weeks of age, so maybe the window considered is not the best one to demonstrate this thesis.
- For IF staining especially the Cy5 ghrelin in Fig.6 in MDX, it seems a lot of background. Is it possible to see the secondary only? Moreover, even the microvessel should be more detectable in the WT (fig.6). it seems that all the MDX related staining have been overexposed.
- Is it possible to have the dot plot in the graphs? Representing the number of field and images analysed?
- It is stated in the material and methods that the whole picture has been taken but there is no whole picture in the manuscript.
- In which way the authors do the direct correlation of increase GHSR expression with worst cardiac outcome (LVEF)?
- It seems that the mechanism through which grelin admistration could ameliorates muscle “integrity” does not depend from receptor expression.
- In the discussion, stating that the correlation with GHSR expression correlates with cardiac function with such a low N number, it is probably risky.
Minor comments:
- Mdx goes always in italics
- Mdx:utrn has to be written always in the same way
- Line 43 and Line 119 the formatting is not correct
- Line 465 there is a full stop that is not needed after reference 43, and is absent after the word inflammation.
Round 2
Reviewer 3 Report (New Reviewer)
Comments and Suggestions for Authors
I thank the authors for the improvements to the manuscripts and to have clarify all my doubts. The paper is more clear. I have no further comments.
Author Response
We thank the reviewer for their helpful comments in improving the manuscript.
This manuscript is a resubmission of an earlier submission. The following is a list of the peer review reports and author responses from that submission.
Round 1
Reviewer 1 Report
Comments and Suggestions for Authors
The authors present the results of a prospective study demonstrating increased levels of growth hormone secretagogue receptor (GHSR) in the myocardium of mdx:utrn-/- mice exhibiting a severe Duchenne muscular dystrophy phenotype. The authors showed that an inytancrease in GHSR in the hearts of mdx:utrn-/- mice correlates with the severity of pathology development. Further work involves assessing the impact of GHSR ligands on the development of pathology. I have several remarks and comments regarding this work:
1. The strain of mice must be indicated in the title.
Introduction.
2. The authors write that «The majority of DCM patients die within 5 years of becoming symptomatic». This must be confirmed with a reference.
3. The authors write that «Inflammation is a key component of DMD skeletal muscle pathology, as demonstrated in the mdx mouse model of DMD». The authors cite references 5-9. Strictly speaking, only 2 papers from this list represent original research on mdx mice. The authors should be consistent and either rewrite this sentence, citing experimental papers, or focus on a couple of recent review papers on inflammation. All such points should be verified in the rest of the paper.
3. The authors write that «mdx mouse hearts are fibrotic, hypertrophied and exhibit diminished contractility». Changes in electrophysiological parameters of the mdx heart should also be noted. In mdx mice they are observed even earlier, this has been shown in a number of recent studies (for example, in the case of studying the effect of non-immunosuppressive analogues of cyclosporine A, etc.), although in the case of mdx mice the pathology is certainly very mild.
4. The authors should briefly describe the characteristics of mdx:utrn-/- mice. In particular, the purpose of utrophin knockout, this aggravates the pathological phenotype, but this may not be clear to the reader. References should also be provided in this section.
5. The authors write that «ghrelin administration significantly improved heart function and decreased LV remodelling and fibrosis in a mouse model of dilated cardiomyopathy». Specify this model, it is genetic, this is important.
Results and discussion.
6. The authors write that «Our analysis focusSed on the left ventricular myocardium». Correct the word, and also justify the choice of this particular area of ​​the heart, since this is obvious only to a specialist.
7. H&E images need to be magnified further, details are poorly visible. Perhaps the 10x should be removed and another magnification should be used (maybe 60 or 80x).
8. I recommend providing quantitative data, it will be more objective. For example, the number of Pyknotic nuclei per section, foci of aggregation of lymphocytes, etc.
9. Masson's trichrome staining images should also provide quantitative data on the area of ​​fibrosis.
10. Lines 241 and 275. References Sullivan 2019, 2021 need to be corrected.
11. Fig. 5. The contrast of the WT and mdx samples is very different, please align them, it will be more correct. Moreover, the quantitative data do not differ so much, and from the WT photo one can decide that GHSR fluorescence is completely absent.
12. The authors write that "GHSR may be localized in areas other than cardiomyocytes." Where else, any suggestions?
13. Line 379. The authors write that «IL-6 is also elevated in the diaphragm muscle of mdx mice». Point out that this is the most affected muscle in this strain of mice.
14. Line 397. «The mouse model of diabetic cardiomyopathy showed very mild changes in heart function and no signs of inflammation or fibrosis». Provide a reference.
15. The authors also write that «Therefore, we suggest that GHSR may be elevated in late-stage cardiomyopathy, and down-regulated in early-stage cardiomyopathy in mice». Of course, it is desirable to see age-related changes in the parameter being studied. If this is not possible, then this should be clearly stated as a limitation of the study.
16. The authors also write that «Therefore, we suggest that GHSR may be elevated in late-stage cardiomyopathy, and down-regulated in early-stage cardiomyopathy in mice». This is a very bold statement when compared to diabetic cardiomyopathy. The authors did not conduct work at an early stage of the pathology, so I recommend refraining from such assumptions.
17. Regarding the discussion. I encourage the authors to speculate on the role of GHSR signaling in the development of DMD. In particular, in skeletal muscle, GHSR is involved in the modulation of autophagy and mitophagy, as well as mitochondrial function. Disturbances in all these processes are also observed in DMD. How might the observed changes in GHSR levels be related to these disruptions?
18. mdx should be italic
Reviewer 2 Report
Comments and Suggestions for Authors
The manuscript tackles an important aspect of DCM in DMD and potential role of GHSR in DCM. The results obtained are important, but the manuscript lacks the quantitative data that this study needs. Microscopy images are not a good tool for quantification and the data must be corroborated with robust quantification tools like Western Blots or qPCR or Flow cytometry. As major conclusions from this study are drawn from quantification of fluorescence microscopy images, it makes use of other tools for quantification more important.
The background of the images in WT and DMD panels of Figure 5, 7 and 9 look very different. The background also affects the quantification. If the images are used for quantitative analysis, the background intensities must be similar. It is difficult to see the presence of GHSR in the WT panel of these images, although GHSR is bound to be present in WT cells as well. A difference in the expression levels using western blot would be highly beneficial for the study.